# Transcriptomics-Based Study of Immune Genes Associated with Subclinical Mastitis in Bactrian Camels

**DOI:** 10.3390/vetsci12020121

**Published:** 2025-02-02

**Authors:** Wanpeng Ma, Huaibin Yao, Lin Zhang, Yi Zhang, Yan Wang, Wei Wang, Yifan Liu, Xueting Zhao, Panpan Tong, Zhanqiang Su

**Affiliations:** 1College of Veterinary Medicine, Xinjiang Agricultural University, Urumgi 830052, China; 320220066@stu.xjau.edu.cn (W.M.); 320230074@stu.xjau.edu.cn (L.Z.); 120130015@xjau.edu.cn (Y.Z.); 320222746@stu.xjau.edu.cn (Y.W.); 320232873@stu.xjau.edu.cn (W.W.); 320232797@stu.xjau.edu.cn (Y.L.); 320242918@stu.xjau.edu.cn (X.Z.); 120170026@xjau.edu.cn (P.T.); 2Xinjiang Key Laboratory of New Drug Research and Development for Herbivorous Animals, Urumgi 830052, China; 3Xinjiang Laboratory of Special Environmental Microbiology, Institute of Microbiology, Xinjiang Academy of Agricultural Sciences, Urumqi 830091, China; yaohuaibing@stu.xju.edu.cn

**Keywords:** Bactrian camel, mastitis, RNA Seq, immune pathways, gene expression, camel milk

## Abstract

Camels are invaluable to many regions globally, with Bactrian camels specifically providing essential resources such as milk, meat, and hair. Subclinical mastitis, an inflammatory breast disease, significantly impacts the productivity and health of Bactrian camels, yet research on its immune pathways remains scarce. We aimed to address this gap by conducting the first transcriptome analysis of Bactrian camel subclinical mastitis, comparing immune responses between healthy and mastitic Bactrian camels. We believe that our study makes a significant contribution to the literature because our study revealed 1722 differentially expressed genes, including key immune genes like *IL10*, *CCL5*, and *IL1B*, highlighting the cytokine–cytokine receptor interaction pathway’s crucial role in Bactrian camel subclinical mastitis. This novel insight into the immune regulatory mechanisms of subclinical mastitis in Bactrian camels can enhance diagnostic and preventive strategies, benefiting both the camel milk industry and public health.

## 1. Introduction

Camels have significant economic value as a special species in many regions of the world [1]. Currently, there are 35 million camels worldwide, of which 89% are Camelus dromedarius and 11% are Bactrian camels [2]. Camelus dromedarius is mainly distributed in the Arabian Peninsula, Afghanistan, India, Pakistan, and southern and western Africa. Bactrian camels are predominantly found in Mongolia, Kazakhstan, Russia, China, and other regions [3,4], providing essential daily necessities such as camel milk, meat, and hair for many pastoralists in arid areas [5]. Camel milk contains high amounts of minerals and vitamins [6], which confer antibacterial and antidiabetic properties and can improve immunity, prevent disease, reduce the risk of cancer [7], and inhibit angiotensin-converting enzymes [8]. Lactoferrin (cLf) in camel milk can inhibit the growth of various microorganisms [9]; therefore, camel milk is often used for medicinal purposes [10]. As dairy animals, camels are also susceptible to mastitis [11].

Mastitis is an inflammatory disease of the mammary gland tissue, mainly caused by infections and trauma to the mammary gland tissue, it is predominantly clinical and subclinical mastitis, causing camel milk spoilage and resulting in increased treatment and labor costs [12,13,14]. Studies have reported a 52.5% incidence rate of mastitis in Pakistani dromedaries caused by microorganisms such as *Staphylococcus aureus*, *Escherichia coli*, and *Bacillus cereus* [15]. In Saudi Arabian dromedaries, the incidence rate of subclinical mastitis is 44.4%, primarily caused by microorganisms such as *S. aureus* and *Corynebacterium*, while 80% of Sudanese dromedaries’ mastitis cases are caused by *S. aureus* [16,17,18]. Furthermore, high-throughput sequencing of microorganisms in the milk of camels with mastitis revealed that the main bacteria present were *Staphylococcus.spp*, *Streptococcus.spp*, and *Schlegeliella.spp* [19]. Compared to Camelus dromedarius, mastitis in Bactrian camels in Xinjiang, China, is mainly caused by microorganisms such as *E. coli*, *Streptococcus agalactiae*, *S. aureus*, and *Enterococcus faecalis*. This condition significantly impacts the development of the Bactrian camel milk industry and poses a public health risk [20].

When subclinical mastitis occurs in camels, the contents of CD45 (cluster of differentiation), white blood cells, CD172a, and myeloid cells, among others, in camel milk increase, while the number of lymphocytes decreases [21,22]. The development of subclinical mastitis is related to the enhanced phagocytic activity of cells such as phagocytes, macrophages, and granulocytes [23]. Further, CD8 + T cells are more highly expressed in healthy lactating camel mammary glands than in non-lactating healthy camels, whereas MAdCAM-1 is highly expressed in non-lactating healthy camel mammary gland tissue, indicating a close relationship between the transportation of camel mammary gland cells and the intestinal immune system [24]. Additionally, studying the cell population in camel mammary gland tissue through the expression patterns of CD markers and adhesion molecules revealed that lymphocyte transport exhibits mucosal properties [25]. Research on camel immunity has mainly focused on immune organs, immunoglobulins, and intestinal immunity [26,27,28]. However, these studies have paid very little attention to the functioning of the immune system of camels with mastitis, and the transcriptional changes that occur in Bactrian camels with mastitis remain unclear. Therefore, we compared the transcriptome responses in the blood of Bactrian camels with subclinical mastitis and healthy Bactrian camels. This study represents the first transcriptome analysis of Bactrian camel mastitis, with our data providing new insights into the immune research of subclinical mastitis in Bactrian camels.

## 2. Materials and Methods

### 2.1. Experimental Animals and Sample Collection

The Bactrian camels used in this study were all Junggar Bactrian camels from Jimunai County, Xinjiang, China. Jimunai County is in the northern part of Xinjiang Uygur Autonomous Region, Bactrian camels in this area live mainly free range from April to November. They feed on wild forage, such as camel thorn, red willow, white thorn, Calligonum mongolicum, and Peganum harmala. Herders supplement their diet with a small amount of alfalfa and corn flour. The animals live in captivity from November to March of the following year. Herders feed them on wild forage, alfalfa, corn straw, wheat straw, corn flour, and other feeds.

In this study, we chose a breeding farm housing 120 Bactrian camels and employed CMT to screen the milk from all four mammary glands of lactating Bactrian camels, Ultimately, we selected 4-peak CMT-positive and 3-peak CMT-negative Bactrian camels, all aged between 7 and 8 years. These camels shared the same parity (both had two fetuses), lactation period (four months), and feeding environment (primarily fed on alfalfa). We collected 2 mL of jugular vein blood from each camel. The blood was added to EP tubes containing 6 mL of TRIzol and shaken until the blood and TRIzol were completely mixed. After letting the mixture stand for 5 min, it was loaded onto liquid nitrogen and transported to the laboratory for future use [22,29,30].

### 2.2. Experimental Methods

#### 2.2.1. RNA Extraction, Library Construction, and Transcriptome Sequencing

Total RNA was extracted from seven blood samples using the TRIzol reagent (Beijing Dingguo Changsheng Biotechnology Co., Ltd., Beijing, China). The RNA integrity number (RIN) was detected using an Agilent 2100 bioanalyzer. After obtaining qualified total RNA, the mRNA with a polyA tail was enriched using oligo (dT) magnetic beads. Subsequently, the obtained mRNA was randomly interrupted with divalent cations in Fragmentation Buffer (Murray Biotechnology Co., Ltd., Shanghai, China). The first strand of cDNA was synthesized using fragmented mRNA as a template and random oligonucleotides as primers in the M-MuLV reverse transcriptase (Beijing Biotech Co., Ltd., Beijing, China) system. Subsequently, RNA chains were degraded using RNaseH (Beijing Biotech Co., Ltd., Beijing, China). The second strand of cDNA was synthesized using dNTPs as raw materials in the DNA polymerase I (Beijing Biotech Co., Ltd., Beijing, China) system. The purified double-stranded cDNA was repaired at the end, followed by adding an A-tail connected to a sequencing adapter. cDNA of approximately 370–420 bp was screened using AMPure XP beads for polymerase chain reaction (PCR) amplification, and the PCR product was purified again using AMPure XP beads to obtain the library. The Agilent 2100 bioanalyzer was used to detect the size, concentration, and quality of the cDNA library, and the final cDNA library was obtained. After confirming the library’s quality, it was sent to Beijing Nuohe Zhiyuan Technology Co., Ltd. (Beijing, China) for Illumina Hiseq 2000 sequencing (Illumina, Inc, San Diego, CA, USA) [31,32].

#### 2.2.2. Data Quality Control and Reference Genome Alignment

The raw data were filtered using the software fastp (version 0.19.7), using fastp-g-q5-u 50-*n* 15-l 150 as the parameters, to ensure the quality and reliability of the data analysis. We excluded reads with adapters, those containing N (N indicates that base information cannot be determined), low-quality reads (reads with base number Qphred ≤ 20 accounting for more than 50% of the entire read length), and finally calculated Q20 and Q30 for clean data. Q20 ≥ 90% and Q30 ≥ 80% indicated high sequencing quality. We used HISAT2 v2.0.5 to align the final clean data quickly and accurately with the reference genome of Bactrian camels (https://www.ncbi.nlm.nih.gov/genome/10741?genome_assembly_id=212083, accessed on 5 April 2023). We obtained the localization information of reads on the reference genome, and the proportion of successfully aligned sequencing reads generated in the experiment was >70% (total mapped reads or fragments).

#### 2.2.3. Identification of Differentially Expressed Genes

The featureCounts (1.5.0-p3) tool in the Subread software was used to analyze the gene expression levels, calculate the expected number of fragments per kilobase of transcript sequence per million base pairs sequenced (FPKM) of each gene based on its length, and calculate the reading mapped to that gene. Based on the analysis of gene expression levels, Pearson correlation analysis (under ideal experimental conditions, the square of the Pearson correlation coefficient (R^2^) is greater than 0.92; however, in specific experiments, R^2^ is generally required to be greater than 8). Using linear algebra computational methods, the obtained gene variables underwent dimensionality reduction and principal component extraction, followed by principal component analysis (PCA). A principal component analysis graph is plotted with log2 (FPKM + 1) as the vertical axis to display the distribution of gene expression levels in different samples. All related graphics are drawn using the NovoMagic cloud platform (https://magic.novogene.com/customer/main#/lo-GinNew. accessed on 7 August 2023).

To explore the differences in gene expression between CMT-positive and CMT-negative Bactrian camels, the DESeq2 software (1.20.0) was used to analyze the differential expression between Bactrian camels with subclinical mastitis and normal Bactrian camels. Benjamin and Hochberg’s methods were used to adjust the obtained *p*-values to control the false detection rate. Through DESeq2, it was found that genes with adjusted *p*-values < 0.05 were assigned as differentially expressed. Therefore, differentially expressed genes (DEGS) were screened using |log2 (FoldChange)| ≥ 1 and padj ≤ 0.05 as the standard.

#### 2.2.4. Functional Enrichment and Protein-Protein Interaction Network Analysis of Differentially Expressed Genes

The NovoMagic Cloud Platform was utilized to conduct Gene Ontology (GO) functional enrichment analysis and Kyoto Encyclopedia of Genes and Genomes (KEGG) pathway enrichment analysis on differentially expressed gene sets. A cut-off of *p* < 0.05 was used to screen significant functions and pathways, which were classified into molecular functions, biological processes, and cellular components. Significant immune-related DEGS and related gene functions were identified through GO enrichment analysis by enriching signal pathways through KEGG. The immune-related genes in DEGS were compared to the STRING protein interaction database, with Camelidae selected as the reference species. Homology with known proteins was used to obtain the interaction relationship between differentially expressed gene-coding proteins. The immune core (hub) gene with the highest connectivity in the PPI network was identified using Cytoscape (version 3.4.0) [33].

#### 2.2.5. Quantitative Reverse Transcription Polymerase Chain Reaction Analysis

Six DEGS (*IL10, ILIB, TNFRSF1B, IL12RB1, CCL5,* and *IFNG*) were randomly selected. Gene mRNA sequences were downloaded from the NCBI website, and β-actin was selected as an internal reference gene. Primers were designed using the Primer-BLAST online tool on the NCBI website (Table 1) and synthesized by Shanghai Shenggong Biotechnology Co., Ltd. (Shanghai, China). RNA was extracted using the method described in 2.2.1, and reverse transcription was performed. The reaction system for RT-qPCR was 20 uL in total, consisting of the following: 10 μL of AceQ Universal SYBR qPCR Master Mix, 0.4 uL each of upstream and downstream primers, 6.7 uL of ddH2O, and 2.5 uL of cDNA. The reaction conditions were 95 °C for 5 min, followed by 40 cycles of 95 °C for 10 s and 60 °C for 30 s. An ABI real-time PCR instrument (model: StepOne Plus) was used to detect the relative expression level of genes, with technical replicates for each sample. The relative expression level of genes was calculated using the 2^−ΔΔCT^ method, and the relative expression level was analyzed using *t*-tests. Finally, GraphPad Prism 8.0.2 was used for plotting [33,34].

## 3. Results

### 3.1. RNA-Sequencing Reads and Mapping to the Reference Genome

After filtering the raw data, 46.67 G clean reads were obtained, with a Q20% greater than 97% and Q30% more significant than 93% (Table 2), indicating high-quality data from this transcriptome sequencing. The alignment rate of each sample ranged from 88.93% to 91.01% when these clean reads were compared to the Bactrian camel reference genome. The success rate of read alignment to the genome was >70%, suggesting a complete assembly of the reference genome. Additionally, the alignment rate to the unique position of the reference genome ranged from 85.8% to 89.23% (Table 3 and Appendix A).

### 3.2. Analysis of Gene Expression Levels

The box plot shows that the similarity between the median, maximum, and minimum values of genes among the seven samples was relatively high (Figure 1A). The density distribution map shows highly consistent and uniform expression distributions of the seven samples (Figure 1B), which indicates that the data obtained from sequencing in this experiment are relatively reliable. The Pearson correlation analysis results show a strong correlation between biological replicates in each sample (Figure 2), and the inter-group samples are scattered in the PCA plot, indicating significant differences in expression between samples (Figure 3).

### 3.3. Gene Differential Expression Analysis

A total of 17,202 expressed genes were found in the seven samples, and 1722 DEGS were ultimately screened, including 1061 upregulated genes and 661 downregulated genes (Figure 4). The clustering heatmap of DEGS shows differences in gene expression levels between the two groups of samples (Figure 5).

### 3.4. GO Function and KEGG Signaling Pathway Enrichment Analysis Results of Differentially Expressed Genes

GO functional enrichment analysis was performed on 1722 DEGS obtained through screening, and a total of 433 DEGS were detected in the GO database, enriched in 644 trees. Among these, 236 were enriched in MF, 98 in CC, and 310 in BP. Enrichment was found on the 10 GO pathways with the highest expression in each process (Figure 6), and the proportion of extracellular region part and cytoskeletal part in the cell components was the highest. Further, the proportion of signal transmitter activity, signal receptor activity, and transmembrane signal receptor activity in the molecular functional group was the highest. Among the biological process components, the proportion of the G-protein-coupled receptor signaling pathway was the highest, followed by internal signal transmission, and it contained immune response pathways (a total of 15 immune genes enriched, including 8 upregulated genes and 7 downregulated genes) related to immunity. The KEGG enrichment analysis of the DEGS revealed that 668 DEGS were enriched in 309 KEGG pathways. The most significant 20 KEGG pathways were selected, and immune-related pathways were found in the most significant 20 KEGG pathways (Figure 7), including cytokine–cytokine receptor interaction (a total of 44 immunodifferentially expressed genes were enriched, of which 36 were upregulated and 8 were downregulated), and complex and coagulation cascades. The pathways of natural killer (NK) cell-mediated cytotoxicity, T helper type 17 cell differentiation, and Th1 and Th2 cell differentiation, among others, were significantly dominated by terms related to immune response and inflammation.

### 3.5. Protein–Protein Interaction Network of Differentially Expressed Genes

Using the STRING database and Cytoscape software analysis, changes in core DEGS such as *AARHGEF26, ARHGEF10L, VAV3, PLEKHG4B, TSC2, HCST,* and *NET1* were more expressed in Bactrian camels that developed subclinical mastitis. Furthermore, we found that core DEGS related to immunity such as *IL10, CCL5, IL1B, OSM, TNFRSF1B, IL7, CCR3, IL4R, IFNG*, and *IL18* were highly associated with central node proteins (Figure 8). The functions and enrichment pathways of these key hub proteins further revealed changes in immune genes related to Bactrian camels with subclinical mastitis.

### 3.6. RT-qPCR Validation of Differentially Expressed Genes

Six core DEGS including *IL10, ILIB, TNFRSF1B, IL12RB1, CCL5,* and IFNG were randomly selected for RT-qPCR validation, as shown in Figure 9. The RT-qPCR expression patterns of the six core DEGS were consistent with the RNA sequencing (RNA Seq) results, further demonstrating the reliability of RNA Seq.

## 4. Discussion

Mastitis is the most common disease in the global dairy industry [35]. For instance, in cows alone, there are more than 200 microorganisms that cause mastitis [36]. Once these microorganisms infect cows, the economic loss per cow annually amounts to EUR 230 [37], which escalates with increasing parity [38]. Currently, the incidence rates of cow mastitis in the United States, Finland, India, and China are 24.4%, 16.1%, 86.32%, and 8.1%, respectively [39,40,41,42]. In contrast, the incidence rate of mastitis in Bactrian camels in China has reached 26% [20], surpassing that of cows in most countries and emerging as a significant disease affecting the development of the camel milk industry [43]. Camels are highly adaptable to extreme desert ecosystems and have stronger disease resistance than other animals in the same region, prompting extensive research into their unique immune systems [44,45]. As a mammal capable of producing functional homodimeric immunoglobins, camel serum contains a large number of H-chain-only Igg. The antigen-binding fragments of these unique hcAbs contain only a single structural domain. These microscopic antigen-binding fragments may play important roles after microbial stimulation of the udder [46]; the major histocompatibility complex region of the camel possesses many large clusters of immune-related genes, such as lymphocyte antigens, complement factors, and tumor necrosis factors, which may play important roles in mammary gland immunity [47]. However, there is limited research on the immune pathways and responses related to camel mastitis [25,48]. Therefore, this study used RNA Seq to investigate the main immune pathways and genes involved in camels after mastitis, identifying 1722 DEGS, including 1061 upregulated and 661 downregulated genes. GO enrichment found that the pathways related to the immune response in Bactrian camels after mastitis were mainly concentrated in biological processes, further demonstrating that the systemic response after breast infection with pathogens involves biological processes [49]. The breast immune system comprises various bodily fluids and cytokines, which help eliminate breast pathogens [50].

The functions of genes involved in subclinical mastitis in Bactrian camels were further investigated through core DEGS identified by a PPI network analysis. It has been proven that ARHGEF26 has the ability to promote angiogenesis and interact with angiogenetic factors and pathways [51]. ARHGEF10L overexpression can promote cell proliferation and migration and reduce cell apoptosis [52]. As a Rho guanine nucleotide exchange factor (Rho GEF), PLEKHG4B plays a crucial role in the formation of epithelial junctions [53]. Vav3 and HCST play a role in the development of human breast cancer [54,55]. The low expression of TSC2 is related to the poor prognosis of breast cancer [56]. The expression of NET 1 can promote the proliferation and differentiation of normal breast epithelial cells [57]. It is predicted that these genes will play a vital role in the development of subclinical mastitis in Bactrian camels. Furthermore, a protein interaction network analysis revealed *IL10, CCL5, IL1B, LCK, IL18, CXCR4, IL7, OSM, TNFRSF1B, IFNG,* and *PLCG1*, among others, to be the main core immune genes. These core genes mainly consist of cytokines such as interleukin, interferon, tumor necrosis factor, and chemokines. This suggests that the cytokine–cytokine receptor interaction pathway may play an important role in subclinical mastitis in Bactrian camels. In contrast to Bactrian camels, bovine mastitis is mainly caused by inflammation, immune response, and endocytosis, followed by interactions between cytokines and cytokine receptors [58].

When the mammary glands of camels are infected with microorganisms, the first step is innate immune system recognition, which in turn triggers persistent adaptive immunity [24,59]. The mammary gland recognizes pathogen-associated molecular patterns (PAMPs) through pattern recognition receptors (PRRs) to initiate innate immune and inflammatory responses [60]. PAMPs and PRRs are associated with non-specific immunity [24], while dendritic cells (DCs) are associated with humoral immunity due to their role as antigen-presenting cells [61]. During microbial infection, breast cells produce a set of cytokines that induce lymphocytes to generate another set of cytokines, triggering a robust inflammatory response and activating local and systemic inflammatory responses [60]. These cytokines participate in many aspects of immune and inflammatory responses, such as innate immunity, adaptive immunity, and expression of adhesion molecules (AMs) [43,62]. AM expression can determine the migration of white blood cells from the blood to the breast. AMs such as selectins, intermediate proteins, and immunoglobulins can regulate signal transmission between immune cells and control the movement of white blood cells to tissues [63]. Local immunity in the breast mainly relies on neutrophils to control infection. The production of interleukin-17 (IL-17) can enhance immunity in the breast [64]. IL-17 is produced by T helper type 17 cells and neutrophils [65], highlighting the important role of T helper type 17 cell differentiation in camel breast immunity. Neutrophils play an important role in innate immunity by killing bacteria through phagocytosis, which is crucial for effectively clearing bacteria and resisting breast infection [66].

Moreover, in some Gram-negative bacterial infections of mastitis, bacterial lipopolysaccharides (LPSs) can directly interact with neutrophils through CD14 expressed on the cell surface [67]. LPSs also promote cytokine production and initiate inflammatory responses [68]. Lymphocytes are transported into the breast under the influence of AMs, facilitating their migration to the breast tissue [25]. The abundant production of neutrophils and lymphocytes may significantly increase somatic cells in the breast, activating the adaptive immune response of the breast and the whole body [23].

After mastitis occurs, tissues and inflammatory cells stimulate the expression and secretion of chemokines [64,69], which accelerates the binding of chemokines to G-protein conjugates (GPCRs), achieving immune monitoring [70]. CCL5, known as the “Type I IFN chemokine characteristic”, attracts NK cells, monocytes, and activated lymphocytes to gather in the breast [71]. NK cells come into contact with various microorganisms and are activated by mediators released by antigen-presenting cells. Once activated, NK cells produce large amounts of IFNG, supporting the development of a Th1 partial response by placing NK cells at the interface between innate and adaptive immunity [72]. An increase in IFNG can improve the clearance rate of bacteria and the resistance of the breast, reducing inflammation production [58,73]. Simultaneously, activated NK cells promote the maturation of DC, and the PPR produced on DC induces inflammation and adaptive immunity in the body by detecting conserved PAMPs [24]. This may also lead to significant enrichment of pathways, such as the G-protein-coupled receptor signaling pathway, Th1 and Th2 cell differentiation, and NK-mediated cytotoxicity, in Bactrian camels with mastitis.

The core genes enriched in this study included the SRC family kinase (SKF) member LCK, the first molecule activated after the T-cell antigen receptor (TCR) participation. Activating LCK signaling can promote T-cell activation and improve T-cell immune responses to pathogens [74,75]. However, it is worth noting that LCK is downregulated in several major pathways after KEGG enrichment, which may be related to the formation of chronic mastitis. Some studies have shown that the severity of mastitis and differences in the pathogens of mastitis are related to differences in the oxidation products of infected breasts [76], and a lack of cytokines in the breast can lead to chronic mastitis [77].

Camel milk may play an important role in the immunity of Bactrian camel mammary glands [22]. It contains abundant antibacterial proteins, such as peptidoglycan recognition protein (PGRP), an essential antibacterial protein that attaches to bacterial lipopolysaccharides (LPSs) or lipoteichoic acids (LTAs) [78]. PGRP plays a vital role in preventing bacterial adhesion and reproduction, and it can inhibit inflammatory cytokines and tumor necrosis factor-α (TNF-α) [10,79]. Lactoferrin has been shown to reduce mortality by interacting with interleukin-6 (IL-6) [79,80]. Additionally, lactoferrin reduces iron availability to pathogens and exhibits antibacterial effects, effectively binding bacteria to mucosal surfaces and inhibiting bacteria proliferation. It is also considered a primary defense against mucosal infections. During the mastitis assessment using the CMT, the author observed strong positive milk on the previous day’s test. On the second day, the milk was determined to be normal, which may result from the unique adaptive immunity of Bactrian camel mammary glands and the mutual influence of the camel milk [81]. Due to the absence of a threshold for somatic cell counting in Bactrian camels [22], this study did not measure the somatic cell count in camel milk. Accurate identification of subclinical mastitis is challenging when relying solely on CMT detection and clinical symptoms. This limitation is clearly visible in the principal component analysis, where despite the dispersion of samples across groups, some subclinically infected samples appear close to healthy ones. At the same time, the number of samples collected in this study was relatively limited. Although DEGS and pathways associated with subclinical mastitis in Bactrian camels have been uncovered, further efforts in collecting samples and enhancing research are still necessary for diagnosing and preventing subclinical mastitis in Bactrian camels at the molecular level, or for breeding Bactrian camels resistant to subclinical mastitis in the future.

## 5. Conclusions

In China, the increasing demand for camel milk has driven the rapid development of Bactrian camel farming, yet Bactrian camel mastitis has a major impact on milk quality. This study used transcriptomics to investigate the immune response of Bactrian camels during subclinical mastitis, identifying 1722 DEGS, including 1061 upregulated and 661 downregulated genes. Functional enrichment analysis revealed significant enrichment in immune response, G-protein-coupled receptor signaling pathways, and cytokine–cytokine receptor interaction pathways. DEGS such as *IL10, CCL5, IL1B, OSM, TNFRSF1B, IL7,* and *CCR3*, among others, implicated in these pathways, may play a crucial role in the development of subclinical mastitis in Bactrian camels, though further analysis is needed to explore their potential functions. The present study sheds light on the DEGS and pathways associated with subclinical mastitis in Bactrian camels, providing insights into its immunoregulatory mechanisms.

## Figures and Tables

**Figure 1 vetsci-12-00121-f001:**
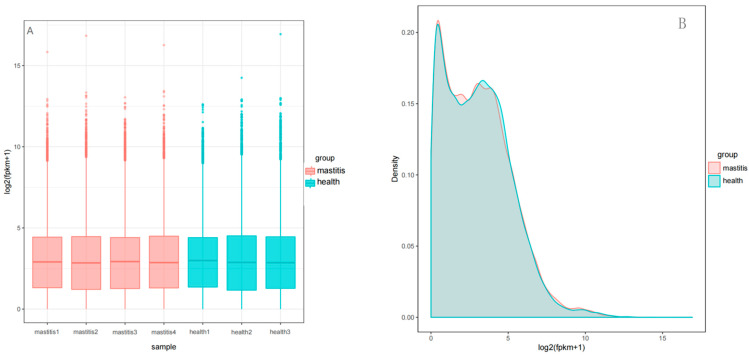
Box plots (**A**) and density distributions (**B**) of gene expression levels in subclinical mastitis Bactrian camel and healthy Bactrian camel samples. The horizontal axis in (**A**) represents the sample name, the vertical axis represents log2 (FPKM + 1). The horizontal axis in (**B**) represents log2 (FPKM + 1), and the vertical axis represents the density of sample expression.

**Figure 2 vetsci-12-00121-f002:**
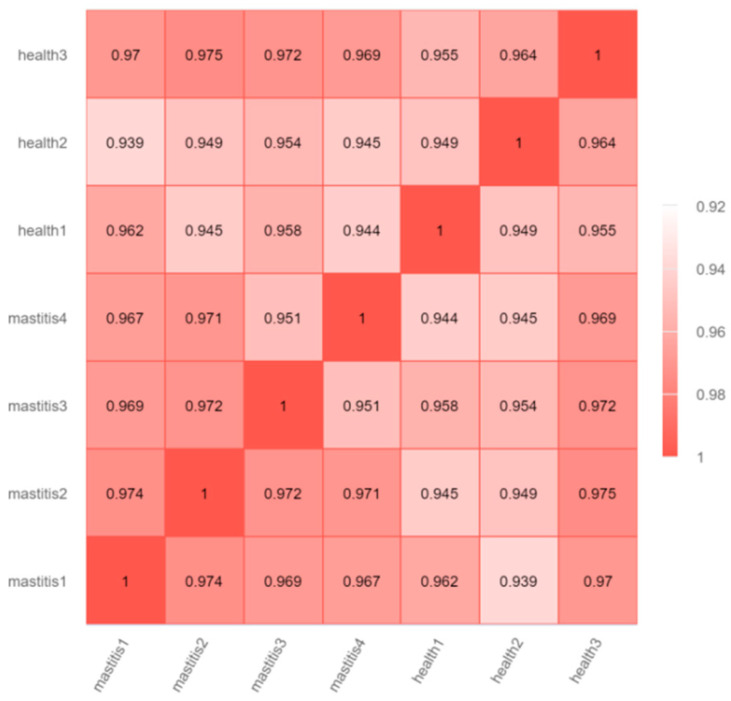
Heat map of correlation between subclinical mastitis Bactrian camel and healthy Bactrian camel samples; the horizontal and vertical coordinates represent the squares of the correlation coefficients in each sample.

**Figure 3 vetsci-12-00121-f003:**
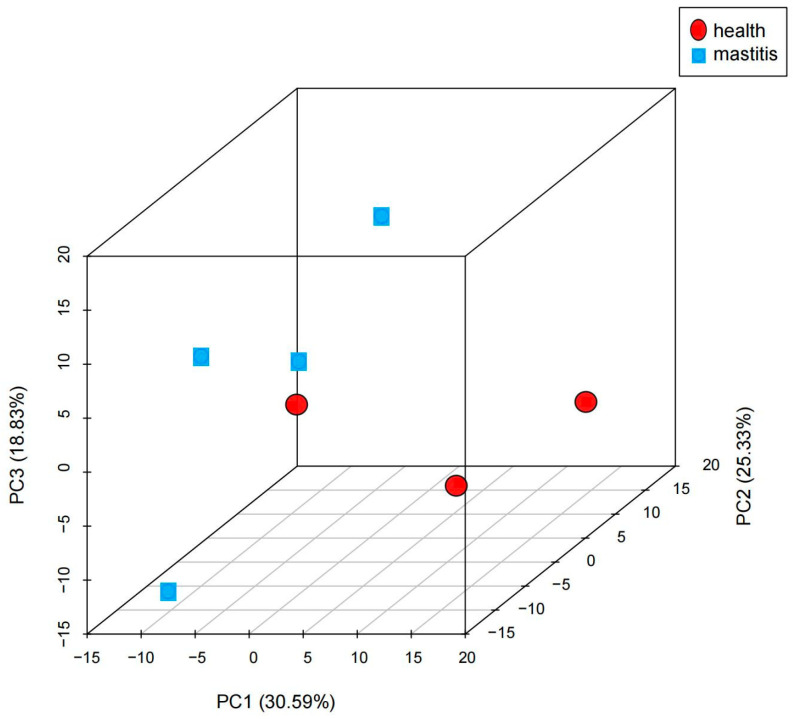
Principal component analysis results of subclinical mastitis Bactrian camel and healthy Bactrian camel samples; the horizontal axis represents the first principal component, and vertical axis represents the second principal component.

**Figure 4 vetsci-12-00121-f004:**
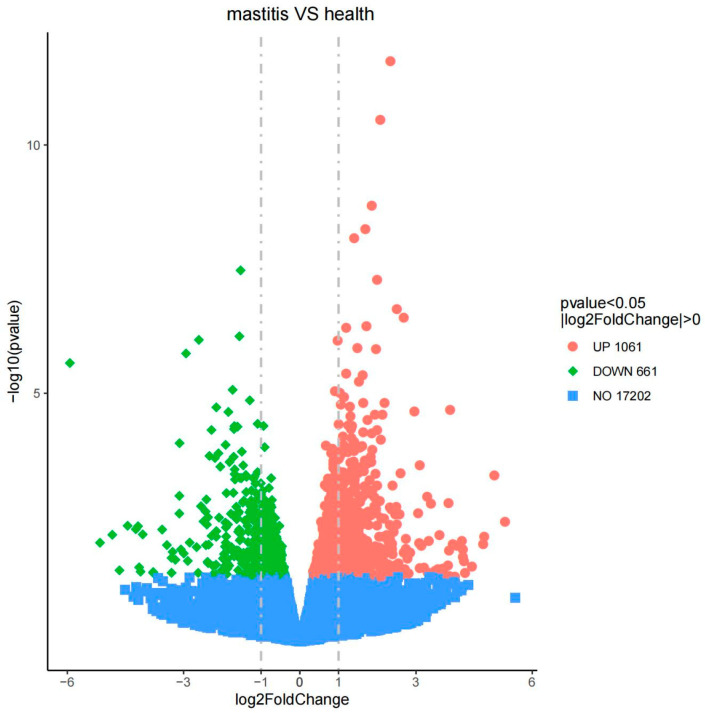
Volcano map of differentially expressed genes between subclinical mastitis Bactrian camels and healthy Bactrian camels; the horizontal axis in the figure is the log2 FoldChange value, and the vertical axis is -log10 *p*-value. The blue dashed line represents the threshold line for differential gene screening criteria.

**Figure 5 vetsci-12-00121-f005:**
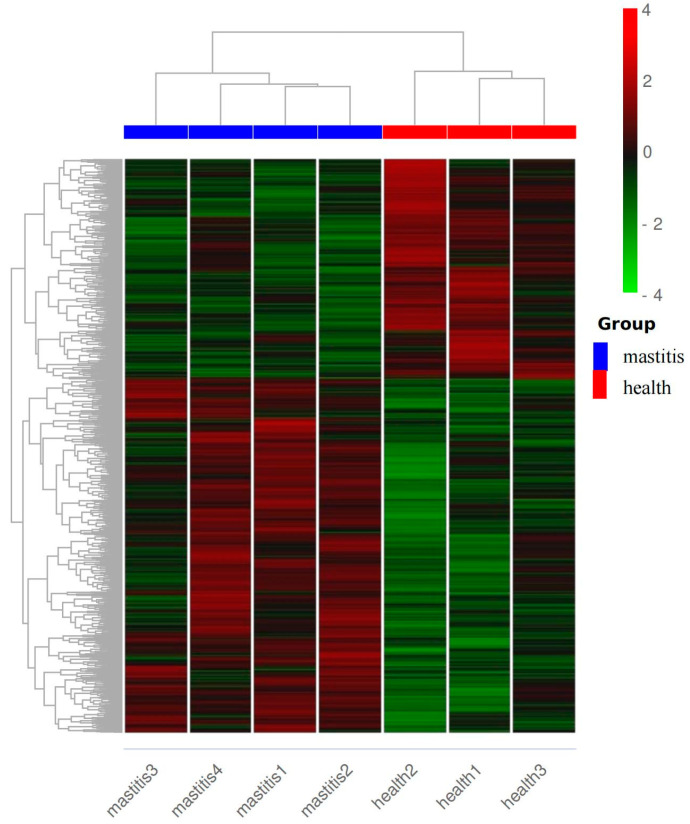
Cluster heatmap of differentially expressed genes between subclinical mastitis Bactrian camels and healthy Bactrian camels. The horizontal axis represents the sample name, and vertical axis represents the normalized value of the differential gene FPKM. The redder the color, the higher the expression level, while the greener the color, the lower the expression level.

**Figure 6 vetsci-12-00121-f006:**
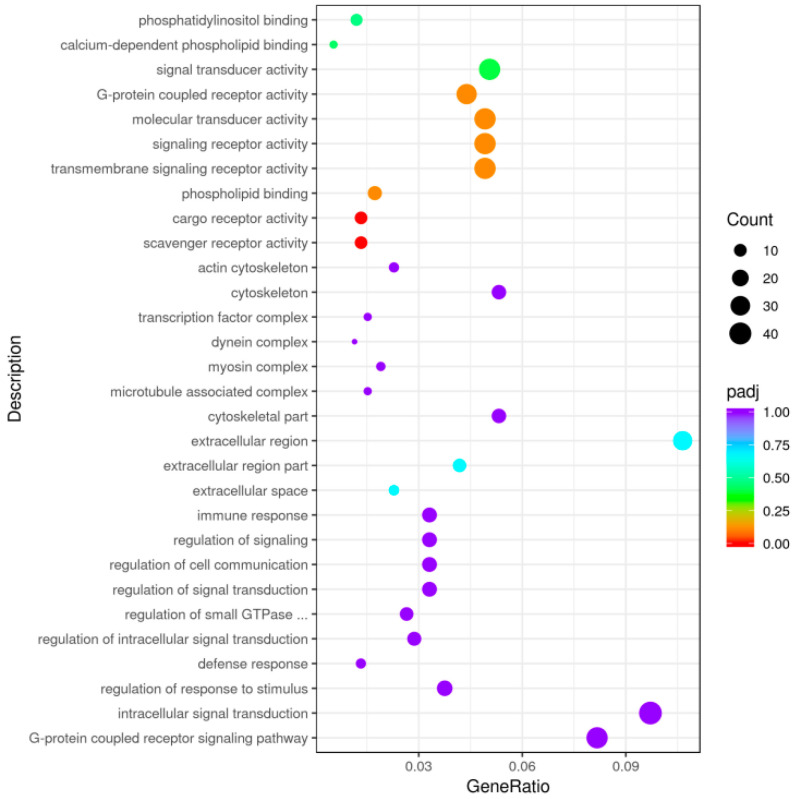
Scatterplot of GO enrichment analysis; the horizontal coordinate is the ratio of the number of differential genes annotated to the GO term to the total number of differential genes, and the vertical coordinate is the GO term.

**Figure 7 vetsci-12-00121-f007:**
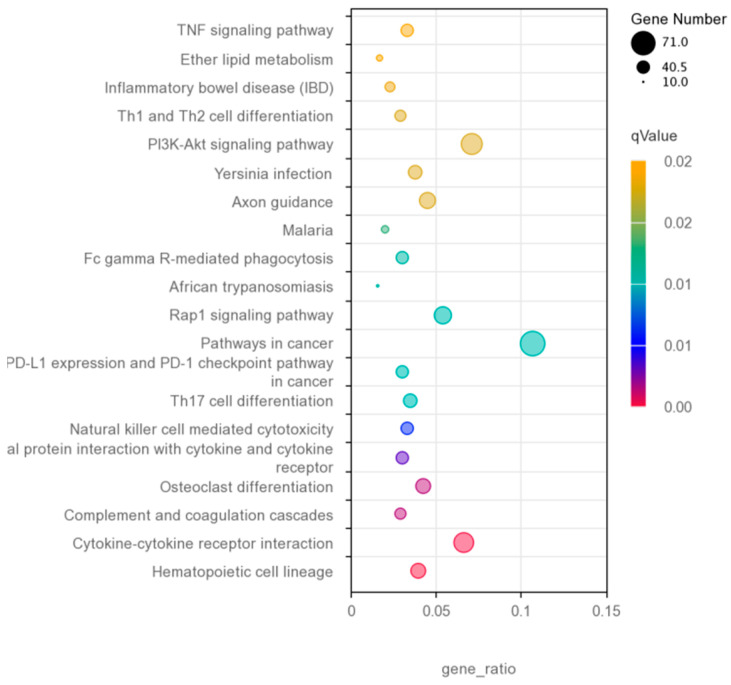
Scatter plot of KEGG enrichment. The horizontal axis represents the ratio of the number of differentially expressed genes annotated on the GO term to the total number of differentially expressed genes, while the vertical axis represents the GO term. The color of the dots in the figure indicates the degree of enrichment, and the redder the color, the more obvious the enrichment. The size of the dot indicates the number of enriched genes, and the larger the dot, the more genes are enriched. GO, gene ontology; KEGG, Kyoto Encyclopedia of Genes and Genomes.

**Figure 8 vetsci-12-00121-f008:**
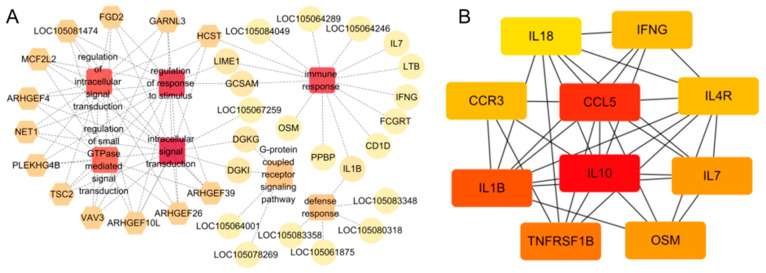
Network analysis of subclinical mastitis-related PPI network and GO pathways (**A**) and immune-related differentially expressed gene PPI network (**B**). (**A**) Non-hub genes are represented by circles, squares indicate pathways, and dark yellow hexagons indicate candidate genes. (**B**) represents the 10 immune genes with the highest connectivity, with darker colors indicating higher connectivity.

**Figure 9 vetsci-12-00121-f009:**
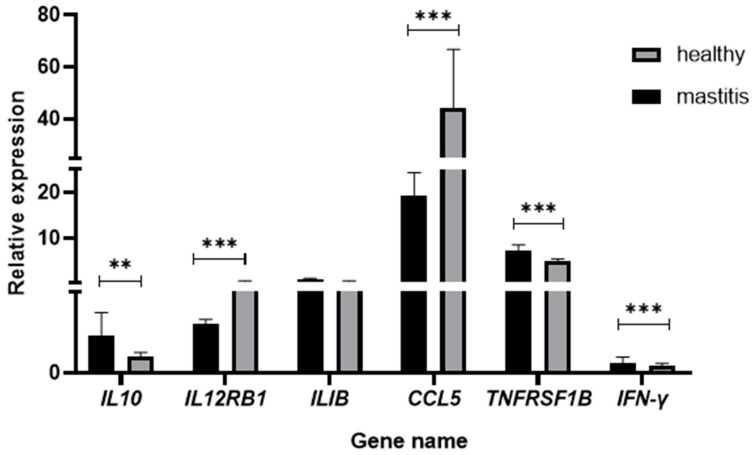
RT-qPCR validation of 6 differentially expressed genes. Note: *** indicates a significant difference, *p* < 0.01, ** significantly different, *p* < 0.05. Compared to normal Bactrian camels, the relative expression levels of *IL10, ILIB*, and *TNFRSF1B* in mastitis Bactrian camels are upregulated, while the relative expression levels of *IL12RB1*, *CCL5*, and *IFNG* are downregulated.

**Table 1 vetsci-12-00121-t001:** Primers for RT-qPCR identification of differentially expressed genes.

Primer Name	Sequence (5′−3′)	Amplification Length (bp)
IL10-F	GCTGTCATCGATTTCTCCCCT	101
IL10-R	CATGGCTTTGTAGACCCCCTT	
CCL5-F	GGCAGCAGTCGTCTTTATCAC	82
CCL5-R	GTTGATGTACTCCCGCACCC	
IL1B-F	CCTCGCACAGGATATGAGCC	73
IL1B-R	CTTGCTGTTGCTTTCGTCCC	
TNFRSF1B-F	AAGTTCCCCAGTTGAAGGGC	75
TNFRSF1B-R	AAGGCTGTCACACCCACAAT	
IL12RB1-F	TCCCCCAAGGTTACCCTGAA	193
IL12RB1-R	CCTGATGTCCACAGTCACCC	
IFNG-F	AATGGCAGCTCCGAGAAACT	86
IFNG-R	CTTATGGCTTTGCGCTGGAC	
ACTIN-F	GATGACGATATTGCTGCGCTC	102
ACTIN-R	CACGATGGAGGGGAAGACAG	

**Table 2 vetsci-12-00121-t002:** Read statistics of subclinical mastitis and healthy Bactrian camel samples before and after transcriptional data filtering.

Sample	RNA Integrity Number	raw_reads	clean_reads	Q20 (%)	Q30 (%)
mastitis1	9.20	46,632,576	45,553,012	97.81	93.81
mastitis2	9.50	47,743,048	46,569,744	97.78	93.73
mastitis3	9.30	46,637,194	45,592,936	97.81	93.86
mastitis4	9.30	46,442,192	45,334,444	97.78	93.77
health1	9.90	45,762,114	44,741,880	97.69	93.44
health2	9.60	41,270,850	40,432,584	97.92	94.07
health3	8.90	45,376,950	44,287,024	97.87	93.96

**Table 3 vetsci-12-00121-t003:** Comparison of transcription data and reference genomes between subclinical mastitis Bactrian camel and healthy Bactrian camel samples.

Sample	total_reads	total_map	unique_map	multi_map
mastitis1	45,553,012	41,127,866(90.29%)	39,694,536(87.14%)	1,433,330(3.15%)
mastitis2	46,569,744	41,899,829(89.97%)	39,955,160(85.8%)	1,944,669(4.18%)
mastitis3	4,5334,444	40,986,165(90.41%)	39,281,294(86.65%)	1,704,871(3.76%)
mastitis4	45,592,936	40,544,094(88.93%)	39,485,372(86.6%)	1,058,722(2.32%)
health1	44,741,880	40,717,538(91.01%)	39,921,667(89.23%)	795,871(1.78%)
health2	40,432,584	36,237,372(89.62%)	35,283,865(87.27%)	953,507(2.36%)
health3	44,287,024	39,502,065(89.2%)	37,693,504(85.11%)	1,808,561(4.08%)

## Data Availability

The datasets presented in this study can be found in online repositories. The names of the repository/repositories and accession number(s) can be found below: https://www.ncbi.nlm.nih.gov/sra/PRJNA1149524, accessed on 27 January 2025.

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
