# Peer review of "Transcriptomics-Based Study of Immune Genes Associated with Subclinical Mastitis in Bactrian Camels"

_vetsci, 2025, doi:10.3390/vetsci12020121_

Round 1

Reviewer 1 Report

Comments and Suggestions for Authors

The Article entitled, "Transcriptomics-based study of immune genes associated with subclincal mastitis in Bactrian camels" has performed the first transcriptomic analysis of blood cells of Bactrian camels and a comparison of subclinical mastitis and healthy samples. This has provided insights into the immunological state of samples from subclinical mastitis cases, something that is novel to the camel mastitis field. This transcriptomic dataset is well described in the article and the raw data is provided in the supplemental materials. This is a well written article with clear novelty.

General Concept Comments:

The methodology is well written and detailed. The data is clearly interpreted and supported by the figures and not overstated. I am grateful that the authors have provided raw data in the supplemental files. The discussion is extensive but lacks discussion about the limited number of samples used, the variability within the data, and what the future plans for this work are. Additionally, I would like increased statistical analysis and statements on the statistical tests performed. 

Specific comments:

Statistics - Which stat test was used for Figure 9, The IFN-y seems highly significant but with overlapping variance. Please can you comment on the statistical approach for this figure and add details to the text?

What statistical analysis was used to support the statement in line 232? Have you performed any clustering analysis of the PCA to determine if the mastitis samples cluster separately from the healthy samples? If not, is there anything else that can explain the statistical clustering like age of the camel or severity of the disease?

Figure 3 - it would be useful to increase the thickness of the samples on the plot, the mastitis samples do not currently print well in black and white.

Figure 4 - it is standard practice to not have a central line on the plot but to show the log2FoldChange cut-offs of -1 and 1 on the plot (cut-offs stated in the methods). Does this change the number of upregulated and downregulated genes? If so, please make those adjustments in the text and data files. 

Discussion - please add in a paragraph that talks about the limitations in sample size of the current study as well as what the future plans for this work could be. It would be useful for you to also comment on the host variation present between each camel, especially in light of a clustering analysis of the PCA. 

Minor comments:

There is an issue in the Simple Summary with Subclinical Mastitis being in capital letter and no space between the previous word and the start of Subclinical.

Line 80 & 81 there appears to be a word missing that would differentiate the two lists of bacteria.

Line 88, typo in myeloid.

Throughout, bacterial names should be in Italics. After the first mention, subsequent mentions can use the shortened version e.g. Staphylococcus aureus and S. aureus.

In section 2.1 it would be useful to include how many animals have been included from each experimental group.

Title for 2.2.4 has a typo for Differentially.

Line 312, should read IL10 rather than L10.

Line 358, and should not be in italics.

Line 386, I'm pretty sure you mean Gram-negative bacteria. Please add in the word Gram

Lines 422 and 423, do you mean to mention viruses here? I assume this is an error and should read bacteria?

In section 3.4 there are abbreviations that it would be helpful for the reader to explain at this point.

Author Response

Dear reviewer:

Thanks for your questions and suggestions. Following your comments, we have carefully re edited the entire manuscript,Sincerely thank you again for your suggestion.

Comments 1: Statistics - Which stat test was used for Figure 9, The IFN-y seems highly significant but with overlapping variance. Please can you comment on the statistical approach for this figure and add details to the text?

Response 1: The relative expression level of genes was calculated using the 2-ΔΔCT method, and the relative expression level was analyzed using t-tests, Finally, GraphPad Prism 8.0.2 was used for plotting.The revised line number is 208-211.

Comments 2: What statistical analysis was used to support the statement in line 232? Have you performed any clustering analysis of the PCA to determine if the mastitis samples cluster separately from the healthy samples? If not, is there anything else that can explain the statistical clustering like age of the camel or severity of the disease?

Response 2: Using linear algebra computational methods, the obtained gene variables undergo dimensionality reduction and principal component extraction, followed by conducting principal component analysis (PCA),A principal component analysis graph is plotted with log2(FPKM+1) as the vertical axis to display the distribution of gene expression levels in different samples.The revised line number is 167-173.

Comments 3: Figure 3 - it would be useful to increase the thickness of the samples on the plot, the mastitis samples do not currently print well in black and white.

Response 3: Regarding the issue with Figure 3, it has been redrawn.

Comments 4: Figure 4 - it is standard practice to not have a central line on the plot but to show the log2FoldChange cut-offs of -1 and 1 on the plot (cut-offs stated in the methods). Does this change the number of upregulated and downregulated genes? If so, please make those adjustments in the text and data files.

Response 4: Figure 4 has been revised to a standard volcano plot as suggested by the reviewer, with the centerline removed and log2FoldChange threshold values of -1 and 1 added.

Comments 5: Discussion - please add in a paragraph that talks about the limitations in sample size of the current study as well as what the future plans for this work could be. It would be useful for you to also comment on the host variation present between each camel, especially in light of a clustering analysis of the PCA.

Response 5: Shortcomings of this study:Due to the absence of a threshold for somatic cell counting in Bactrian camels, this study did not measure the somatic cell count in camel milk, Accurate identification of subclinical mastitis is challenging when relying solely on CMT detection and clinical symptoms, This limitation is clearly visible in the principal component analysis, where despite the dispersion of samples across groups, some subclinically infected samples appear close to healthy ones. At the same time, the number of samples collected this time is relatively limited. Although DEGS and pathways associated with subclinical mastitis in Bactrian camels have been uncovered, further efforts in collecting samples and enhancing research are still necessary for diagnosing and preventing subclinical mastitis in Bactrian camels at the molecular level, or for breeding Bactrian camels resistant to subclinical mastitis in the future. The revised line number is 437-447.

Comments 6: There is an issue in the Simple Summary with Subclinical Mastitis being in capital letter and no space between the previous word and the start of Subclinical.

Response 6: The Subclinical Mastitis in the abstract has been corrected to "subclinical mastitis" with a space added before "Subclinical".

Comments 7: Line 80 & 81 there appears to be a word missing that would differentiate the two lists of bacteria.

Response 7: The genera and species of each bacterium have been italicized, and "spp" has been added after some genera. The revised line number is 76-84.

Comments 8: Line 88, typo in myeloid.

Response 8: The incorrect word "myeloid" has been corrected to "myeloid". The revised line number is 89.

Comments 9: Throughout, bacterial names should be in Italics. After the first mention, subsequent mentions can use the shortened version e.g. Staphylococcus aureus and S. aureus.

Response 9: The bacterial names have been italicized, and multiple occurrences of bacteria have been replaced with their abbreviated forms.

Comments 10: In section 2.1 it would be useful to include how many animals have been included from each experimental group.

Response 10: In Section 2.1, it has been stated that the experimental animals used in this study comprised 4 CMT-positive Bactrian camels and 3 CMT-negative Bactrian camels.The revised line number is 119.

Comments 11: Title for 2.2.4 has a typo for Differentially.

Response 11: The error in the title of section 2.2.4, where “Ifferentially”' was incorrectly spelled, has been corrected to “Differentially”. The revised line number is 183.

Comments 12: Line 312, should read IL10 rather than L10.

Response 12: L10 has been modified to IL10. The revised line number is 198.

Comments 13: Line 358, and should not be in italics.

Response 13: The italicized 'and' has been modified. The revised line number is 364.

Comments 14: Line 386, I'm pretty sure you mean Gram-negative bacteria. Please add in the word Gram

Response 14: The term "negative bacterial" has been revised to “Gram-negative bacteria”. The revised line number is 392.

Comments 15: Lines 422 and 423, do you mean to mention viruses here? I assume this is an error and should read bacteria?

Response 15: The virus in the original text has been substituted with bacteria. The revised line number is 432.

Comments 16: In section 3.4 there are abbreviations that it would be helpful for the reader to explain at this point.

Response 16: Regarding the abbreviations GO and KEGG mentioned in section 3.4, the full names of the relevant abbreviations have already been updated in section 2.2.4, hence they were not updated in section 3.4.

Reviewer 2 Report

Comments and Suggestions for Authors

There are some grammatical errors in the drafting of the manuscript (e.g. line 72, “It” should be “it”), the authors need to re-read the work carefully.
The genus and species of each bacterium should be indicated in italics, and when only the genus is indicated, then it should be followed by the words “spp.”
The geographical coordinates presented at lines 109-110 are not necessary.
The description of the environment and flora present in the region of origin of the animals is not necessary (lines 111-121)
Line 122, this is a statement that should not be placed in M&M and therefore should be removed.
Line 123-124, this is not necessary. This paragraph regarding animals and sampling appears to be somewhat off-topic. The authors should describe only the animals involved in the study, not the region.
Lines 124-126: How many animals? From how many farms? What then is the calving order? What about the stage of lactation? What about their diet (not a description of what is normally done in the region, but the specific diet of these enrolled animals)? How was subclinical mastitis diagnosed? Did it involve the whole udder or a hemimammary?
Lines 127-128, in M&M it is not necessary to explain why you did something and bibliographic referings should be restricted only to cases of indication of cut-off values or procedures.
What material was used in the collection? Which blood tubes?
How many positive and how many negative animals?
Therefore, was no bacteriological examination performed?
How was the transport of the samples performed?
Line 136, did you extract RNA from 7 samples because that is the total number of animals? If not, why only 7?
Statistical analysis data are broken up between paragraphs of M&M. This form prevents proper evaluation of the approach used (that it is impossible in its present form) as it is scattershot as well as incorrect. Authors should indicate as the last sub-paragraph (which will be 2.3) of M&M all statistical analysis performed with special reference to the choice of sample size, distribution of data, and statistical models used.
For all software, it should be indicated who produced it, city and state, and if such software is online, the site should also be indicated.

In its present form, the manuscript is difficult to read and understand given the lack of appropriate description of the animals enrolled, the relevant experimental design, and the statistical analyses performed. In addition, a paragraph in the Discussion regarding the limitations of the study is missing.

Author Response

Dear reviewer:

Thanks for your questions and suggestions. Following your comments, we have carefully re edited the entire manuscript,Sincerely thank you again for your suggestion.

Comments 1: There are some grammatical errors in the drafting of the manuscript (e.g. line 72, “It” should be “it”), the authors need to re-read the work carefully.

Response 1: "It" has been changed to "it". The revised line number is 73.

Comments 2: The genus and species of each bacterium should be indicated in italics, and when only the genus is indicated, then it should be followed by the words “spp.

Response 2: The genera and species of each bacterium have been italicized, and "spp" has been added after some genera. The revised line number is 76-84.

Comments 3: The geographical coordinates presented at lines 109-110 are not necessary.

Response 3: Geographic coordinates have been removed.

Comments 4: The description of the environment and flora present in the region of origin of the animals is not necessary (lines 111-121)

Response 4: The environment and flora of the place of origin have been deleted.

Comments 5: Line 122, this is a statement that should not be placed in M&M and therefore should be removed.

Response 5: Has been deleted,The large-scale breeding of Bactrian camel is a new breeding industry in Xinjiang, China. The age of a lactating female camel is 5 to 10 years.

Comments 6: Line 123-124, this is not necessary. This paragraph regarding animals and sampling appears to be somewhat off-topic. The authors should describe only the animals involved in the study, not the region. 

Comments 7: Lines 124-126: How many animals? From how many farms? What then is the calving order? What about the stage of lactation? What about their diet (not a description of what is normally done in the region, but the specific diet of these enrolled animals)? How was subclinical mastitis diagnosed? Did it involve the whole udder or a hemimammary?

Comments 9: What material was used in the collection? Which blood tubes? 

Comments 10: How many positive and how many negative animals? 

Comments 12: How was the transport of the samples performed?

Response (6,7,9,10,12): It has been modified to:In this study, we chose a breeding farm housing 120 Bactrian camels and employed CMT to screen the milk from all four mammary glands of lactating Bactrian camels,Ultimately, we selected 4-peak CMT-positive and 3-peak CMT-negative Bactrian camels, all aged between 7 and 8 years,These camels shared the same parity (both had two fetuses), lactation period (four months), and feeding environment (primarily fed on alfalfa). We collected 2mL of jugular vein blood from each camel,The blood was added to EP tubes containing 6 mL Trizol and shaken until the blood and Trizol were completely mixed. After letting the mixture stand for 5 min, it was loaded onto liquid nitrogen and transported to the laboratory for future use.The revised line number is 117-125.

Comments 8: Lines 127-128, in M&M it is not necessary to explain why you did something and bibliographic referings should be restricted only to cases of indication of cut-off values or procedures.

Response 8: Has been deleted,When animals develop mastitis, the integrity of the mammary vascular endothelium is lost, causing systemic reactions.

Comments 11: Therefore, was no bacteriological examination performed?

Response 11: This study primarily explored the immune regulatory mechanism of subclinical mastitis in Bactrian camels, and as a result, no bacteriological examination was conducted.

Comments 13: Line 136, did you extract RNA from 7 samples because that is the total number of animals? If not, why only 7?

Response 13: RNA was extracted from 7 samples in this study, with the specific extraction amounts detailed in Table 2.

Comments 14: Statistical analysis data are broken up between paragraphs of M&M. This form prevents proper evaluation of the approach used (that it is impossible in its present form) as it is scattershot as well as incorrect. Authors should indicate as the last sub-paragraph (which will be 2.3) of M&M all statistical analysis performed with special reference to the choice of sample size, distribution of data, and statistical models used.

Comments 15: For all software, it should be indicated who produced it, city and state, and if such software is online, the site should also be indicated.

Response (14,15): We have modified the statistical analysis in M&M and added the NovoMagic cloud platform to draw relevant graphics.The revised line number is 173.

Comments 16:In its present form, the manuscript is difficult to read and understand given the lack of appropriate description of the animals enrolled, the relevant experimental design, and the statistical analyses performed. In addition, a paragraph in the Discussion regarding the limitations of the study is missing.

Response 16:Shortcomings of this study:Due to the absence of a threshold for somatic cell counting in Bactrian camels, this study did not measure the somatic cell count in camel milk, Accurate identification of subclinical mastitis is challenging when relying solely on CMT detection and clinical symptoms, This limitation is clearly visible in the principal component analysis, where despite the dispersion of samples across groups, some subclinically infected samples appear close to healthy ones. At the same time, the number of samples collected this time is relatively limited. Although DEGS and pathways associated with subclinical mastitis in Bactrian camels have been uncovered, further efforts in collecting samples and enhancing research are still necessary for diagnosing and preventing subclinical mastitis in Bactrian camels at the molecular level, or for breeding Bactrian camels resistant to subclinical mastitis in the future.The revised line number is 437-447.
